# An integrated framework for quantifying immune-tumour interactions in a 3D co-culture model

Gheed Al-Hity [1], FengWei Yang [2], Eduard Campillo-Funollet [3], Andrew E. Greenstein [4], Hazel Hunt [4], Myrthe Mampay [1], Haya Intabli [1], Marta Falcinelli [1], Anotida Madzvamuse [5✉], Chandrasekhar Venkataraman [5✉] & Melanie S. Flint [1✉]

Investigational in vitro models that reflect the complexity of the interaction between the immune system and tumours are limited and difficult to establish. Herein, we present a platform to study the tumour-immune interaction using a co-culture between cancer spheroids and activated immune cells. An algorithm was developed for analysis of confocal images of the co-culture to evaluate the following quantitatively; immune cell infiltration, spheroid roundness and spheroid growth. As a proof of concept, the effect of the glucocorticoid stress hormone, cortisol was tested on 66CL4 co-culture model. Results were comparable to 66CL4 syngeneic in vivo mouse model undergoing psychological stress. Furthermore, administration of glucocorticoid receptor antagonists demonstrated the use of this model to determine the effect of treatments on the immune-tumour interplay. In conclusion, we provide a method of quantifying the interaction between the immune system and cancer, which can become a screening tool in immunotherapy design.

[1] School of Pharmacy and Biomolecular sciences, University of Brighton, Centre for Stress and Age-related Diseases, Moulsecoomb, Brighton BN2, 4GJ, UK. [2] Department of Chemical and Process Engineering, University of Surrey, Surrey, UK. [3] Genome Damage and Stability Centre, University of Sussex, Falmer, Brighton BN1 9QH, UK. [4] Corcept Therapeutics, 149 Commonwealth Drive, Menlo Park, California 94025, United States. [5] School of Mathematical and Physical Sciences, University of Sussex, Department of Mathematics, Falmer, Brighton BN1 9QH, UK. ✉email: A.Madzvamuse@sussex.ac.uk; C.Venkataraman@sussex.ac.uk; M.Flint@brighton.ac.uk

The immune system plays a major role in the body's anti-tumour response. T lymphocytes, a subpopulation of the immune cells, have the capability to infiltrate solid tumours. It is widely accepted that the infiltration of immune cells can impact the tumour and act as a prognostic indicator[1–6]. Researchers are proposing that solid tumours can be classified on the extent of the T cell infiltration, a process called immunoscoring. With the advances in today's technology, pathologists are able to evaluate the immunoscore on tumour tissues extracted from patients and predict prognosis[7,8]; however, reliable non-invasive in vitro models are lacking.

The low success rate in today's clinical trials can be attributed to the lack of translation between in vitro and in vivo research. The optimal culture model would replicate the pathophysiological disease-specific microenvironment. Three-dimensional (3D) models exhibit better resemblance of the tumour microenvironment than the monolayer two-dimensional (2D) models, as they better simulate the solid tumour mass. 3D culture models such as spheroids, provide fundamental information for better understanding of cancer and developing anticancer drugs[9,10]. Spheroid models also offer a more suitable platform for the crosstalk and interactions between tumour cells with other types of cells[11]. Moreover, 3D cultures have been shown to be optimal for studying tumour responses, specifically, to immune stimulation[12]. Generating 3D spheroid cultures using the ultra-low-attachment microplates allows for reproducibility in spheroids' size and shape[13,14]. The use of 3D culture models was specifically recommended for use in breast cancer research[15].

To date, immune-tumour interactions have been assessed using subjective visual comparisons[16]. Quantitative image analysis algorithms are essential, to remove this subjectivity and to allow for robust statistically significant conclusions. The need for quantitative approaches is widely accepted in the drug discovery process and has been suggested and supported by many oncologists as well as governmental agencies to substantially advance the power of clinical studies[17]. The reason behind this multiplanar trend is that the integration of quantitative approaches in cancer models combined with high-performance computing has shown robust results in many different examples[18] as it allows for greater levels of reproducibility, accuracy, and rigour while often offering automation and thereby reducing the labour costs.

Mathematical and computational approaches have been used to provide a great advantage in understanding tumour and vascular growth in response to therapy[19,20], aid in assessing receptor expression[21] and serve as diagnostic tool[22]. Mathematical analysis can also be utilised on cancer models to predict tumour responses to immune infiltration as well as the efficacy of immunotherapies. It is highly recommended by other scientists in the field to develop immune-efficient pre-clinical models[23].

We developed a unified framework for investigating immune-tumour interactions consisting of a co-culture model of immune cells and breast cancer spheroids to study infiltration. We also developed, implemented, and tested two independent quantitative image analysis algorithms designed to measure infiltration levels. Our results show that the immune cells can infiltrate the spheroids and disrupt their structure. We tested the framework by applying it to study the effect of the glucocorticoid, cortisol, on immune infiltration. We showed that stress decreases immune infiltration into tumour spheroids. This framework can be used to screen for different cytokines, and this could help in proposing a mechanism by which the immune cells interact with the tumour. An important aspect of the work is that the two (entirely independent) algorithms for analysing the imaging data led to consistent conclusions.

## Results

**Morphology, optimisation, and viability of spheroids.** In order to establish the co-culture, spheroids from two murine triple negative breast cancer cell lines (66CL4 and 4T1) were grown using ultra low attachment round bottom flasks. It took 4 days for the spheroids to fully form (Fig. 1 A1–A6), where the area and roundness increased over time. The seeding density over the 4 days of experiment duration was optimised as the changes in spheroids should be attributed to the immune cells' infiltration and not to the spheroids themselves. Therefore, both cell lines were seeded at different densities. The seeding density with the largest area and roundness, and with the least variation over the 4 days was chosen. For 66CL4 cells, the seeding density was 1000 cells/well (Fig. 1 B1), while for the 4T1, the seeding density was 1500 cells/well (Fig. 1 B2). The morphology of the spheroids from 66CL4 is substantially different from that of the 4T1; however, both have the capability to form spheroids. The viability of both types of spheroids was measured before the co-culture using the RealTime-Glo 3D viability assay and over 96 h (the co-culture time period). There was no significant loss in viability at 96 h compared to viability at 0 h (Fig. 1C).

**Activated immune cells can infiltrate breast cancer spheroids.** To assess whether the activated immune cells can infiltrate the tumour spheroids, immune cells (splenocytes) isolated from female BALB/c mice were analysed by flow cytometry to determine the expression of the T lymphocyte, CD3, and T-cell activation CD69 surface markers. The cells were stained with the 7AAD viability dye, anti-mouse anti CD3 PE and anti CD69 APC antibodies (Fig. 5A). Splenocytes were split into 3 groups; splenocytes activated with PHORBOL 12-MYRISTATE 13-ACETATE (PMA) and ionomycin for 3 h, 24 h or left without activation (inactivated). Among the 3 groups, there was no significant difference in the number of CD3-positive cells; however, there was a significant increase in the number of CD69-positive cells between inactivated and activated for 3 h ($p = 0.008$) and 24 h ($p = 0.0002$). CD3 and CD69 markers were determined following the gating scheme provided in Supplementary Figure 5. To determine if the activation affected immune cell infiltration, 66CL4 breast cancer spheroids were co-cultured with splenocytes from three different activation settings. Both spheroids and splenocytes were stained with lipophilic tracers, Dio 7778 and Dil D7777, respectively (spheroids with green and immune cells with red) and imaged by phase contrast and confocal microscopy (Fig. 2B). Based on flow cytometric analysis and visual assessment, inactivated and activated splenocytes contained similar numbers of $CD3^+$ cells, whereas activated immune cells consisted of significantly more $CD69^+$ cells. Moreover, when inactivated cells were co-cultured with breast cancer cells, the immune cells were not easily detected due to the lack of red fluorescence in the confocal images and lack of infiltration. Both splenocytes that were activated for 3 and 24 h had significantly higher numbers of $CD69^+$ cells. However, immune cells that were activated for 3 h were easier to identify with lower background levels than the immune cells activated for 24 h; therefore, we chose 3 h for optimal, quantifiable image analysis.

**Restraint stress decreases lymphocytes infiltration into 66CL4 mammary tumours.** In order to validate whether the model resembles the in vivo setting, an in vivo experiment was carried out where female BALB/c mice were injected with 66CL4 cells into the fourth mammary fat pad. Two weeks post injection mice were randomised into a stress group ($n = 6$) and a non-stress group ($n = 6$). Mice from the stress group were subjected to 2 hr of daily

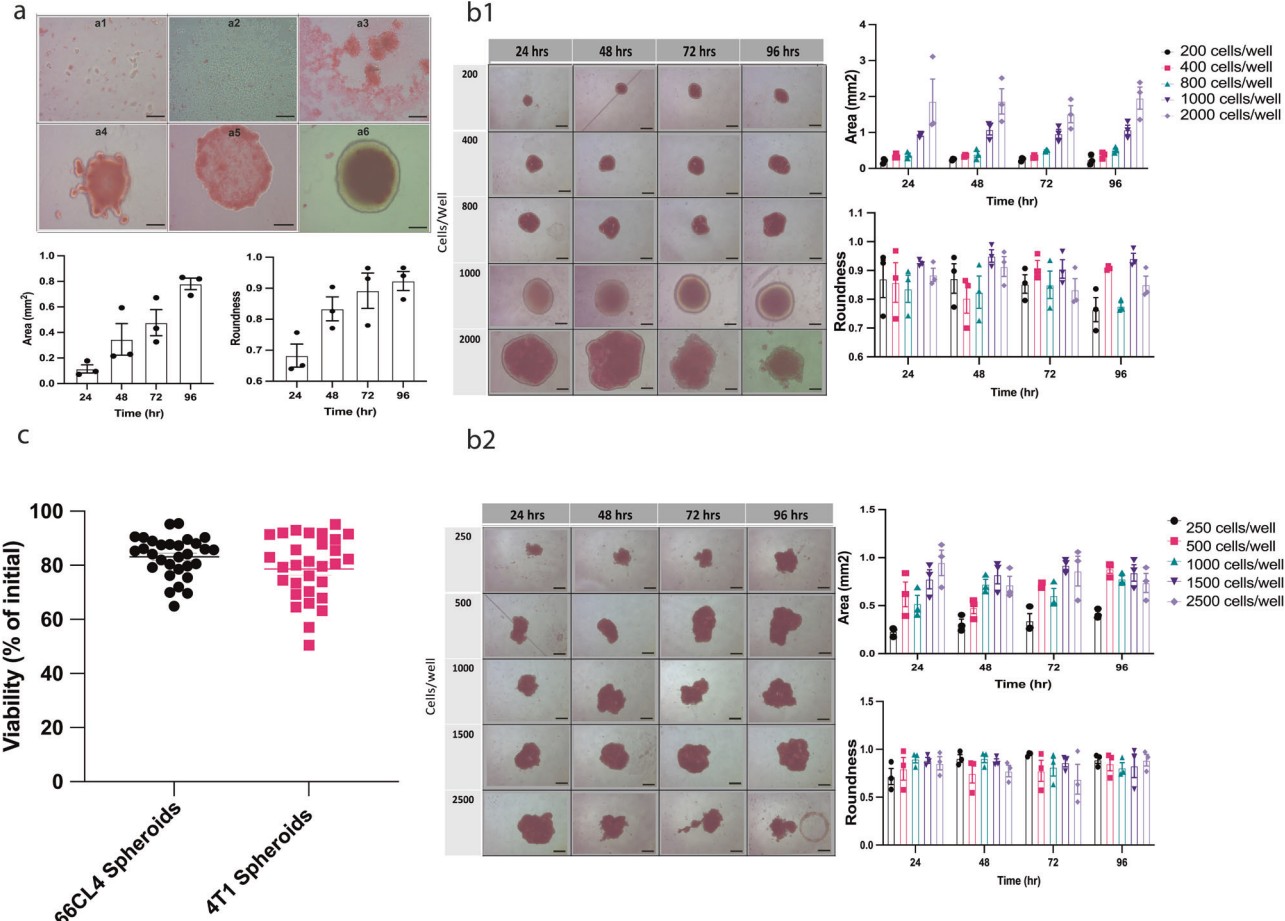

**Fig. 1 Morphology, seeding density and viability of spheroids. a** Phase contrast microscope images showing the visual stages of spheroid generation using ultra-low attachment plates of the murine triple negative breast cancer 66CL4 cell line, from 0 to 96 h: (**a1**) 0 h, (**a2**) 12 h, (**a3**) 24 h, (**a4**) 48 h, (**a5**) 72 h, and (**a6**) 96 h. The projected area and roundness of 66CL4 spheroids for 96 h increased from 0.2 to 0.8 mm², while roundness from 0.7 to 0.9. **b1** 66CL4 spheroid assembly. Representative phase contrast microscope images of spheroids seeded from different number of cells ranging from 200 to 2000 over 4 days after full spheroid generation. The highest level of uniformity in shape (roundness) and size was observed in the spheroids seeded from 1000 cells/well. **b2** 4T1 spheroid assembly. Representative phase contrast microscope images of spheroids seeded from different number of cells ranging from 250 to 2500 cells/well over 4 days after full spheroid generation. The highest level of uniformity is observed in the spheroids seeded from 1500 cells/ well. The scale bar represents 250 μm. All images were taken at objective ×10. Measurements were acquired using Image J software. **c** Cell viability of 3D spheroids (*n* = 30 biologically independent spheroids) from 66CL4 and 4T1 cells. Luminescence was measured over 4 days post culturing cells into spheroids; at time zero of spheroid formation. Readings at 96 h were normalised as a percentage of the initial reading at 0 h.

restraint stress for two weeks; this stress paradigm is known to significantly elevate glucocorticoid levels and affect metatstasis[24]. Mammary tumour weight and volume were measured and then tumours were collected and paraffin-embedded for immunohistochemical analysis. As expected, and although stressed tumours appeared smaller, primary tumour weight and volume were not significantly different between stressed and non stressed groups[25]. Tumour sections were fluorescently labelled with the T-cell marker *CD3ε* and nuclear stain DAPI. Analysis of *CD3* expression in the mammary tumours of stressed and non stressed mice was done using CellProfiler, where the percentage of *CD3*-positive cells was calculated compared to the total number of cells (%CD3 positive/total cells). The amount of T-cell infiltration inside the mammary tumours of stressed mice was significantly reduced compared to the non stressed mice (*p* < 0.001).

**Image analysis of the 3D ex vivo coculture model correlates with an in vivo model**. To test whether stress decreases infiltration levels in our co-culture model in a similar manner to our in vivo experiment (Fig. 3), corticosterone (cortisol) was added to

the co-culture and was compared to a group of spheroids and activated splenocytes only. The co-culture was imaged daily using confocal microscopy and two image analysis approaches we developed for this study were applied on the images to quantify infiltration levels (Fig. 4). Both image analysis approaches sought to measure immune cell infiltration into the spheroid, which we interpret as the number of immune cells (corresponding to red channel intensity) within the tumour spheroid boundary.

Supplementary Figure 1 shows a scatter plot of red and green channel intensity from a random subsample of pixels of the control group that consists of only tumour cells and no immune cells. There is a large amount of red channel fluorescence associated with tumour cells. This presents a major challenge in assessing trafficking levels as it is likely to precipitate spurious evidence of trafficking. We attempted to account for this feature of the data in both of the trafficking measures described below.

The first trafficking index, which we refer to as trafficking index: segmentation (TIS) is computed through a two-stage procedure. First, the tumour masses are segmented in each image (multiple disconnected masses are permissible). Alongside this, we segment connected regions of high red channel intensity

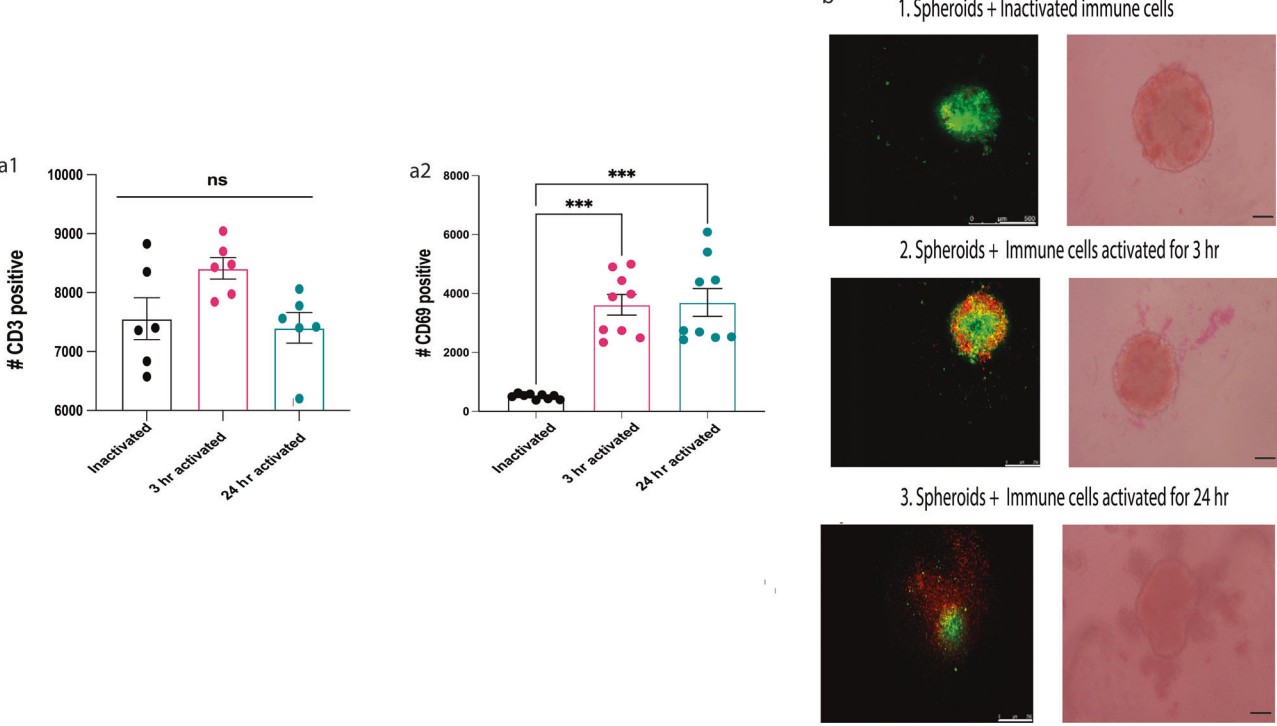

**Fig. 2 PMA and ionomycin activated immune cells isolated from BALB/c (splenocytes) can infiltrate breast cancer spheroids. a1** No significant difference in the number of CD3+ cells in splenocytes between inactivated and activated splenocytes. n = 6 **a2** Activated splenocytes express higher levels of the CD69 activation maker in comparison to the inactivated splenocytes. Statistical significance was calculated using a one-way ANOVA, Kruskal–Wallis test, n = 9 biologically independent animals. Data represents mean ± SEM. *p < 0.05, **p < 0.01, ***p < 0.001. **b** Representative images showing phase contrast as well as confocal microscopic images of spheroids with **b1** inactivated splenocytes, **b2** activated for 3 hr, and **b3** activated for 24 hr. In the confocal images, green represents breast cancer spheroids and red represents splenocytes. The scale bar in the phase-contrast microscopic images represents 250 μm.

which correspond to the immune cells. Details of this segmentation procedure are provided in the "Methods". The TIS is computed as the sum over each pixel that is in both the segmented regions (i.e., tumour and immune cells) of the red channel intensity divided by the green channel intensity. The TIS is the largest in pixels within the tumour (and segmented immune cell region) at which the red channel intensity is high, and the green channel intensity is low, respectively. Hence these pixels unambiguously correspond to immune cells within the tumour region indicating successful trafficking. See Methods for further details of the TIS algorithm.

Second, we developed an alternative independent algorithm to measure trafficking, that does not involve segmentation, and is based on a k-means classifier[26]. The assumption underlying the classification is that the red and green levels of a pixel in a given image characterise the cell type that occupies the region. We trained the k-means classifier on a random subsample of all the available images, aiming to find four classes: tumour cells, immune cells, image background, and colour-saturated regions. The inclusion of the colour-saturated class avoids bias due to the information loss in a saturated pixel. Figure 4b illustrates a sketch of the classification process. The initial training step using a small sample of available images identifies the four classes based on minimising the intraclass variance in the training data. As shown in Fig. 4b, the "colour saturated" class is a large region that includes a number of pixels and indicates the fact that for many pixels, the classifier is unable to simply equate large red channel intensity with high immune cell density. Once trained, the classifier labels each pixel in a given image as belonging to one of the four classes based solely on the red and green intensities of

each pixel. Once each pixel has been classified, we proceed to quantify trafficking. Since we are interested in immune cells inside the tumour spheroid, we count the number of pixels classified as immune cells that are surrounded by pixels classified as tumour cells. In order to obtain a normalised value between 0 and 1, we divide by the total count of pixels surrounded by tumour cells. The resulting statistics yields a number in the interval [0,1] which we refer to as a trafficking index: classification (TIC). See Methods for further details of the TIC algorithm. Since the methods underlying TIC and TIS are completely different, the image regions identified by each method as tumour or immune cells are not completely in agreement. When the spheroid becomes very fragmented or presents an empty region in its interior, the noise reduction from the TIS method tends to overestimate the region corresponding to tumour cells; similarly, the TIC algorithm removes background noise that could otherwise lead to an overestimation of the region corresponding to immune cells. Supplementary Figure 4 illustrates these phenomena. This in turn leads to differences in the trafficking measures that depend strongly on the size of the respective regions.

Figure 5 illustrates the results of both quantitative analyses, where we compute the TIS and TIC, with the statistical conclusions based on both measures being consistent, adding weight to our contention that both algorithms are suited to measuring trafficking levels. The area of the spheroids was quantified, as well as the extent (which is a measure of roundness) to see whether cortisol alters the size or roundness of spheroids. According to TIS and TIC trafficking measures, cortisol significantly decreased the immune cell infiltration to the 66CL4 spheroids (TIS p = 0.0043 and TIC p

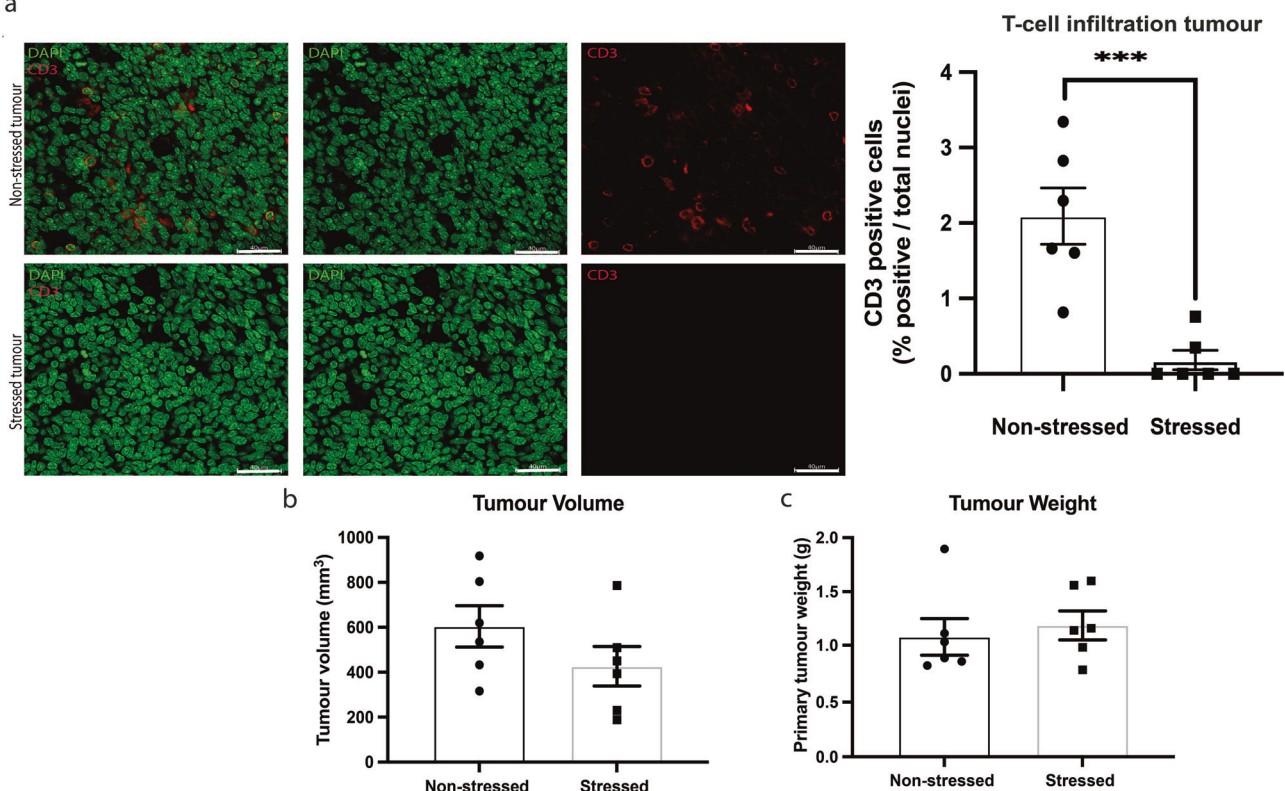

**Fig. 3 Restraint stress significantly decreases T-cell infiltration into 66CL4 mammary tumours but does not affect the primary tumour weight and volume.** Female BALB/c mice were injected with 66CL4 cells into the fourth mammary fatpad. Two weeks post injection mice were randomised into stress ($n = 6$ biologically independent animals) and nonstress ($n = 6$ biologically independent animals) groups. Mice from the stress group were subjected to 2 h of daily restraint stress for two weeks. Mammary tumours were collected and paraffin-embedded for immunohistochemical analysis. **a** Tumour sections were fluorescently labelled with the T-cell marker CD3ε (Cell Signalling, #99940) and nuclear stain DAPI (Sigma, D9542). Analysis of CD3 expression in the mammary tumours of stressed and non-stressed mice was done using CellProfiler, where the percentage of CD3-positive cells was calculated compared to the total number of cells (%CD3$^+$/total cells). The scale bar represents 40 μm. The amount of immune cell–cell infiltration inside the mammary tumours of stressed mice is significantly reduced compared to the non-stressed mice. **b** No significant difference in primary tumour volume between stressed and nonstressed mice. **c** No significant difference in primary tumour weight between stressed and non-stressed Statistical significance was calculated using a two-tailed Mann–Whitney test, data represent mean ± SEM. *$p < 0.05$, **$p < 0.01$, ***$p < 0.001$.

$= 0.0152$). Cortisol did not decrease the spheroids' area, nor did it alter the roundness, correlating with our in vivo study. The co-culture media was analysed for the levels of IFN-γ and IL-10 using ABTS ELISA, cortisol significantly decreased the levels of IFN-γ in the co-culture media ($p = 0.0159$), and significantly increased the levels of IL-10 in the co-culture media ($p = 0.0079$) (Fig. 5). We have also examined the co-culture with and without cortisol at a higher power imaging, and the results were comparable to images taken at a lower magnification (Supplementary Fig. 2).

**Application of the trafficking measures using glucocorticoid receptor antagonists.** After validating our 3D co-culture model both experimentally and computationally, next we describe how this model can be used to assess the effect of drug treatments on trafficking, size, and extent of spheroids by co-culturing with splenocytes and pre-treatment with 3 glucocorticoid receptor antagonists; RU486, and two selective glucocorticoid receptor antagonists; CORT125134 and CORT125281. The co-culture was imaged, and the TIS and TIC trafficking algorithms described above were applied on the images. The images were also analysed for the spheroid area relative to the starting point of the co-culture and the roundness was computed, to see whether the treatments alter tumour spheroid size or disrupt its shape. As shown in Figs. 6 and 7, both visually and quantitatively, each

antagonist affected the immune infiltration, spheroid area and roundness differently. Cortisol significantly decreased immune cell infiltration compared with the untreated spheroids + splenocytes in both 66CL4 (TIS: at 96 h $p = 0.0470$, TIC: at 72 h, $p = 0.0001$) and 4T1 spheroids (TIS: at 48 h, $p = 0.0097$, at 96 h, $p = 0.0094$, TIC: at 72 h, $p = 0.0362$, at 96 h, $p < 0.0001$). CORT125134 reversed the effects of cortisol and significantly increased the immune infiltration in 66CL4 spheroids compared to those cells treated with cortisol (TIS: at 24 h, $p = 0.0032$, at 48 h, $p = 0.0075$). CORT125281 also reversed the effects of cortisol and increased the infiltration of immune cells into 66CL4 spheroids (TIS: at 48 h, $p = 0.0119$, TIC at 72 h, $p = 0.0216$). Also, we have demonstrated that the treatments affect the infiltration and not the retention of the splenocytes' lipophilic tracer (Supplementary Fig. 3).

Spheroid shape was also quantified (Fig. 7). RU486 significantly increased the size of 66CL4 spheroids at 48 h and 96 h compared to spheroids only ($p = 0.0318$ and $p = 0.0443$), the untreated co-culture ($p = 0.0088$ and $p = 0.0072$), CORT125281 ($p = 0.0335$), and at 72 h compared to CORT125134 ($p = 0.0266$), and at 72 and 96 h with spheroids treated with cortisol ($p = 0.0039$, $p = 0.0443$), and the co-culture treated with cortisol ($p = 0.0236$). RU486 also significantly increased the size of 4T1 spheroids at 72 h compared to the untreated co-culture ($p = 0.0067$), the co-culture treated with cortisol ($p = 0.0132$), and spheroids treated

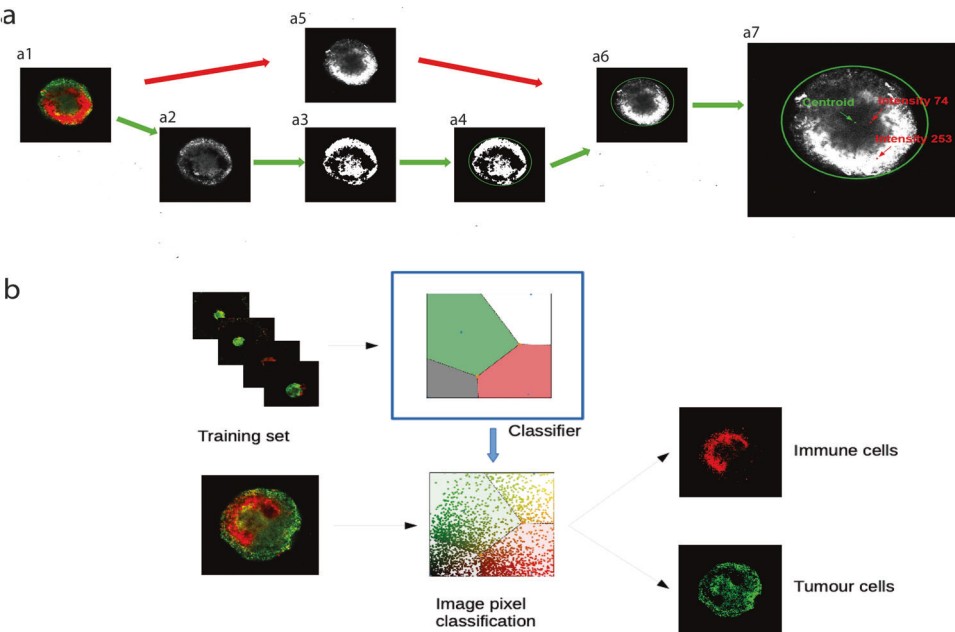

**Fig. 4 Proposed quantitative measures of infiltration. a** Schematic illustrating the two-stage algorithm for computing TIS, the segmentation-based measure of the degree of trafficking. **a1** Raw image data of the green and red channels, where green represents the spheroid cells and red represents the immune cells. **a2** Red channel of the original image. **a3** Green channel of the original image. **a4** and **a5** Enhancement of the red and green channels by noise reduction. **a6** and **a7** Segmented immune cell (**a6**) and tumour (**a7**) regions shaded by red and green intensity, respectively. **a8** Intersection of the segmented tumour and immune cell regions which corresponds to the region where TIS has values greater than zero. **a9** Pixels in the identified region shaded by the value of TIS corresponding to the red/green value. **b** Illustration of the classification of pixels based on a k-means classifier that partitions the red–green space into four disjoint classes. This classification is used to compute the TIC trafficking measure that corresponds to the number of pixels classified as immune cells that are surrounded by pixels that are classified as tumour cells.

with cortisol ($p = 0.0298$). RU486 caused a significant decrease in 4T1 spheroid size at 96 h against spheroids treated with cortisol only ($p = 0.0099$). The immune infiltration alone caused a significant decrease in 4T1 spheroid size against spheroids only ($p = 0.0327$) and against spheroids treated with cortisol ($p = 0.0425$). CORT125134 also significantly reduced the size of 4T1 spheroids against the co-culture treated with cortisol ($p = 0.0291$). The glucocorticoid receptor antagonists also altered the shape, for example RU486 significantly decreased the extent of 66CL4 ($p = 0.0439$) and 4T1 ($p = 0.0031$) spheroids at 24 h and 96 hr, respectively. CORT125134 significantly decreased the extent of 66CL4 ($p = 0.0490$) and 4T1 ($p = 0.0005$) spheroids at 24 h and 96 h. CORT125281 significantly decreased the extent of 66CL4 ($p = 0.0370$) and 4T1 ($p = 0.0311$) spheroids at 24 h and 96 h respectively. Furthermore, the infiltration of immune cells into 4T1 spheroids significantly decreased the extent ($p = 0.0095$) at 96 h.

## Discussion

In this study, we describe a novel framework for studying quantitatively the immune–tumour interaction in breast cancer using a 3D co-culture model and bespoke image analysis algorithms designed to assess infiltration levels (measured through the trafficking index). We first established spheroids from two breast cancer cell lines, optimised the seeding density to create uniformity and proved viability. We showed that different cell lines have varying abilities to form spheroids. 66CL4 cells formed round spheroids, while 4T1 adapted a mass or grape-like shape; these changes are mostly due to the inherent nature of the tumour cells as they were cultured under the same conditions[27]. We chose to use the ultra-low-attachment round-bottom flask method due to its ease of preparation and reproducibility[11]. Second, we used

splenocytes as they reflect the immune system consisting of antigen-presenting cells, as well as, B and T lymphocytes[28] to show that activated immune cells are able to successfully infiltrate the tumour as well as being able to be imaged in this model. High-resolution confocal images were necessary in order to apply the image analysis algorithms used to study the difference between spheroids with and without the immune cells and analyse the images for infiltration, spheroid area, and roundness.

The co-culture model was successfully used to test the effects of the stress hormone cortisol, with a particular emphasis on immune cell infiltration. Immune cell infiltration is fundamental in breast and other cancers and has been strongly linked to response to therapy, as well as patient survival[29,30]. Prior to testing the effect of cortisol on 3D spheroids and immune cell co-cultures, a 66CL4 syngeneic in vivo mouse model was used to test the resemblance of our in vitro model to the in vivo setting. The mice were undergoing restraint stress, which is known to elevate the levels of glucocorticoids in the body[31–33]. The results from our in vitro 3D co-culture model as well as the in vivo experiments showed that stress decreases $CD3^+$ immune infiltration and IFN-γ. This is supported by the literature as many studies have reported that stress negatively affects immune infiltration into tumours[24,25]. In addition, in our model, cortisol did not significantly affect spheroid size. These data are consistent with our in vivo findings, as stress did not affect tumour size or weight and was also supported by other studies[25].

Using two independent image analysis algorithms, we successfully quantified the effects of cortisol on immune infiltration, spheroid size, and shape of the spheroids, as well as the effects of three different glucocorticoid receptor antagonists. The shape assessment is exploratory because the spheroids change shape when immune cells infiltrate, and this may not model a physiological change in tumour shape[34]. The infiltration assessment, in

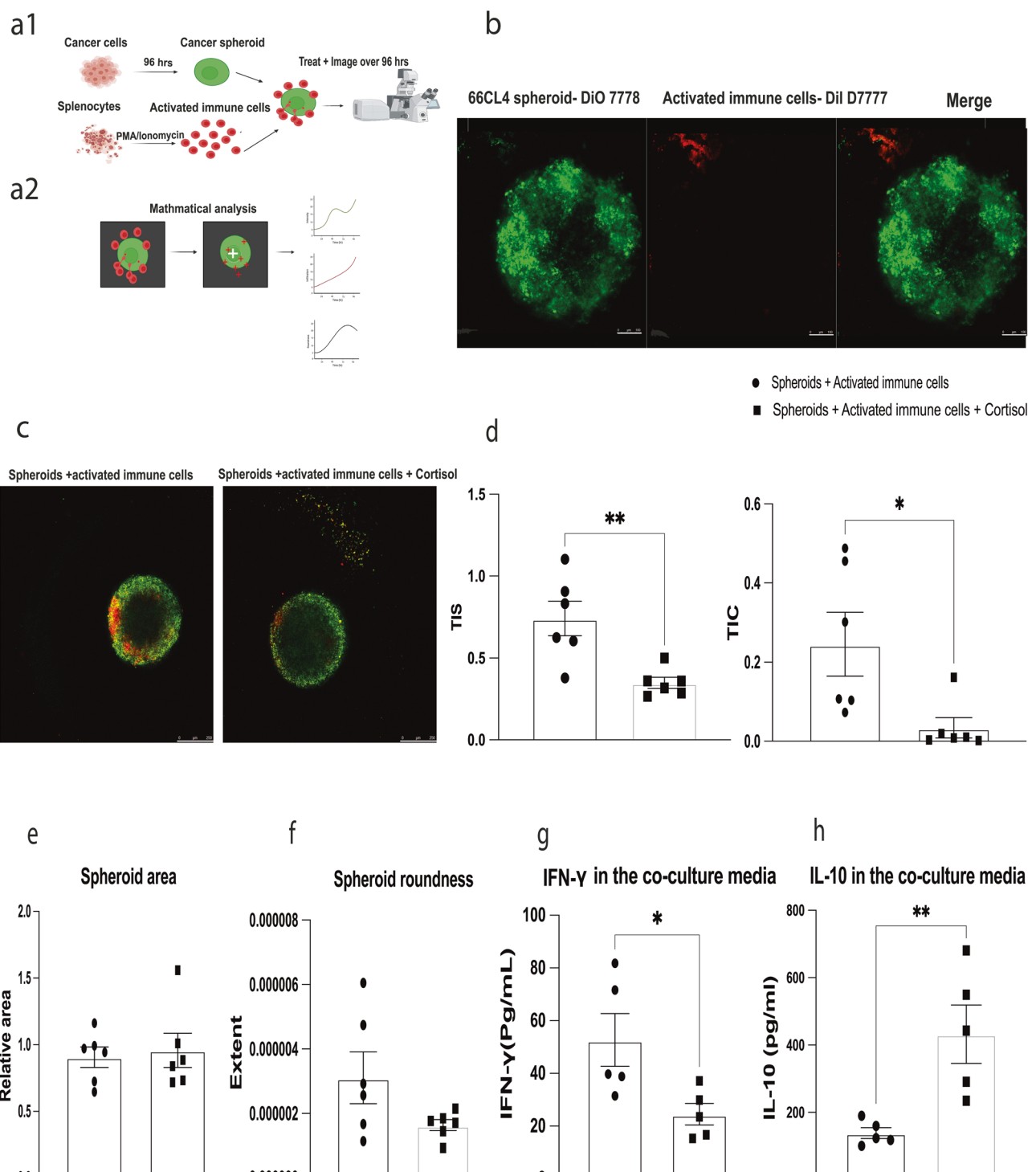

**Fig. 5 Cortisol decreases immune cell infiltration into 66CL4 spheroids but does not affect spheroid area and roundness.** The co-culture model between 66CL4 spheroids and activated immune cells was used to examine the effect of cortisol. **a** Schematic diagram of the co-culture workflow created using Biorender. Initially, spheroids were formed after 96 h post seeding; splenocytes were isolated, activated, and cocultured with spheroids for up to 96 h. The coculture was treated daily with cortisol and imaged daily. **b** Split high-power images of the coculture. Spheroids in green stained with DiO 7778 and immune cells with DiI D7777. The scale bar represents 100 μm. **c** Representative confocal images of the coculture between the 66CL4 spheroids with activated immune cells with or without cortisol, green represents spheroids and red represents immune cells. The scale bar represents 250 μm. **d** Cortisol significantly decreases the infiltration measured using the segmentation-based algorithm (TIS) and the k-means classification-based method (TIC). **e** Cortisol does not affect 66CL4 spheroid area. **f** Cortisol does not alter the 66CL4 roundness. **g** Cortisol significantly decreases the levels of interferon-γ in the coculture media (*n* = 5 biologically independent samples). **h** Cortisol significantly increases the levels of IL-10 in the coculture media (*n* = 5 biologically independent samples). Statistical significance was calculated using a two-tailed Mann–Whitney test. Data represent mean ± SEM. \**p* < 0.05, \*\**p* < 0.01, \*\*\**p* < 0.001.

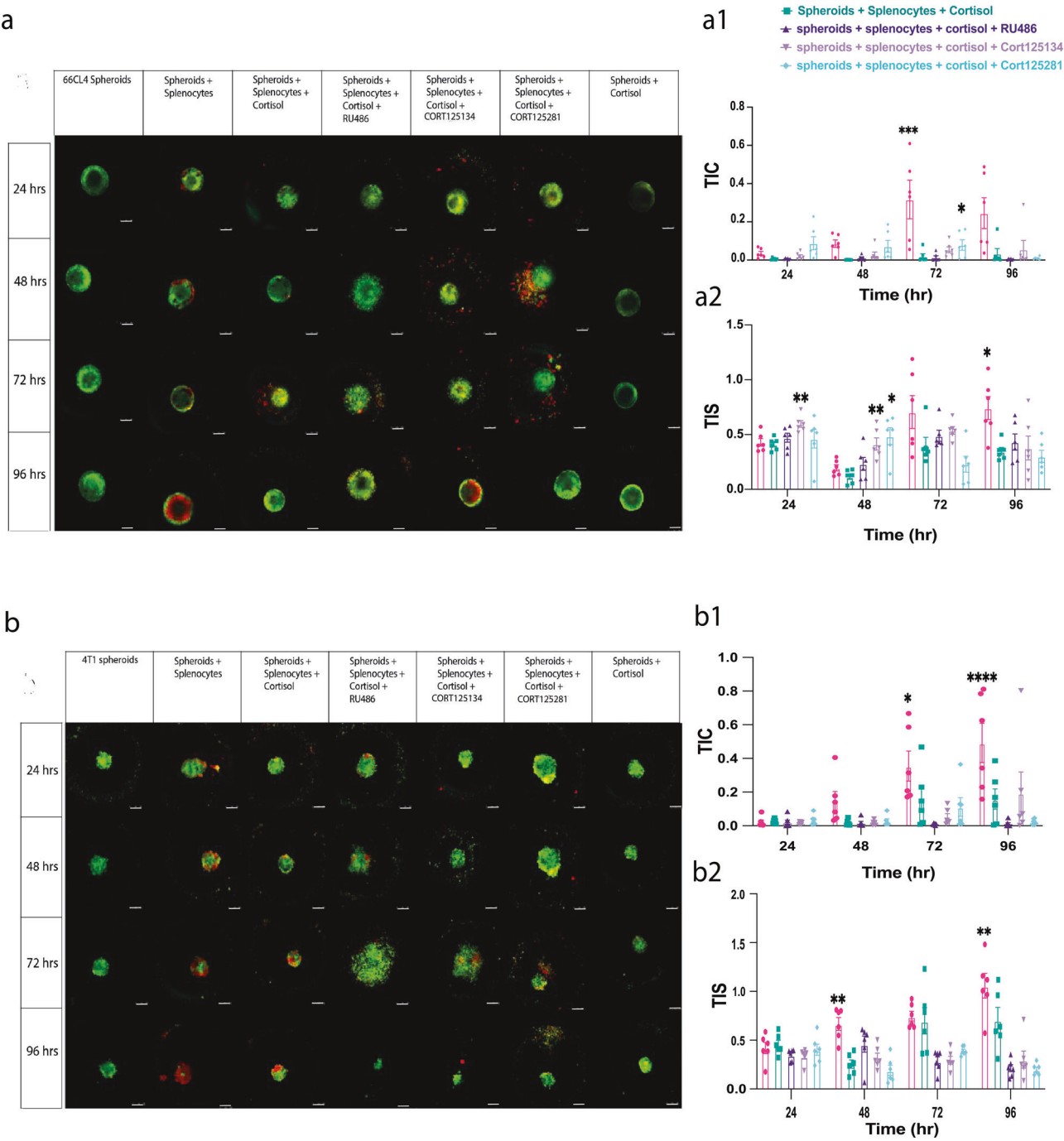

**Fig. 6 Applicability of the model to assess the efficacy of glucocorticoid receptor antagonists. a** Representative images taken using confocal microscopy where green represents the 66CL4 spheroids and red represents immune cells of the co-culture. Spheroids were split into 7 groups: spheroids only, spheroids + splenocytes, spheroids + splenocytes + cortisol, spheroids +splenocytes + cortisol + RU486, spheroids + splenocytes + cortisol + CORT125134, spheroids + splenocytes + cortisol + CORT125281, and spheroids + cortisol. **a1** Image analysis using the TIC trafficking algorithm. **a2** Image analysis using the TIS trafficking algorithm. **b** Representative images of 4T1 spheroids split into 7 groups: spheroids only, spheroids + splenocytes, spheroids + splenocytes + cortisol, spheroids +splenocytes + cortisol + RU486, spheroids + splenocytes + cortisol + CORT125134, spheroids + splenocytes + cortisol + CORT125281, and spheroids + cortisol. **b1** Image analysis using the TIC trafficking algorithm. **b2** Image analysis using the TIS trafficking algorithm. The scale bar represents 250 μm. Significant difference was determined by a two-way ANOVA (treatment + time) followed by Bonferroni post hoc test. Data represent mean ± SEM, $n = 6$ biologically independent spheroids, $*p < 0.05$, $**p < 0.01$, $***p < 0.001$, $****p < 0.0001$.

contrast, directly models a physiological process that impacts the response to drug treatments such as immune modifiers. Although the results between higher and lower magnification were similar, higher magnification for this model can increase light scattering that is considered an inherent limitation of confocal imaging of

thick tissues or multiple layers[35,36] (e.g., immune cells and spheroids).

Our data demonstrate a quantitative model to assess immune cell infiltration. While there are other ways of testing the infiltration of immune cells, they either require more animals or are

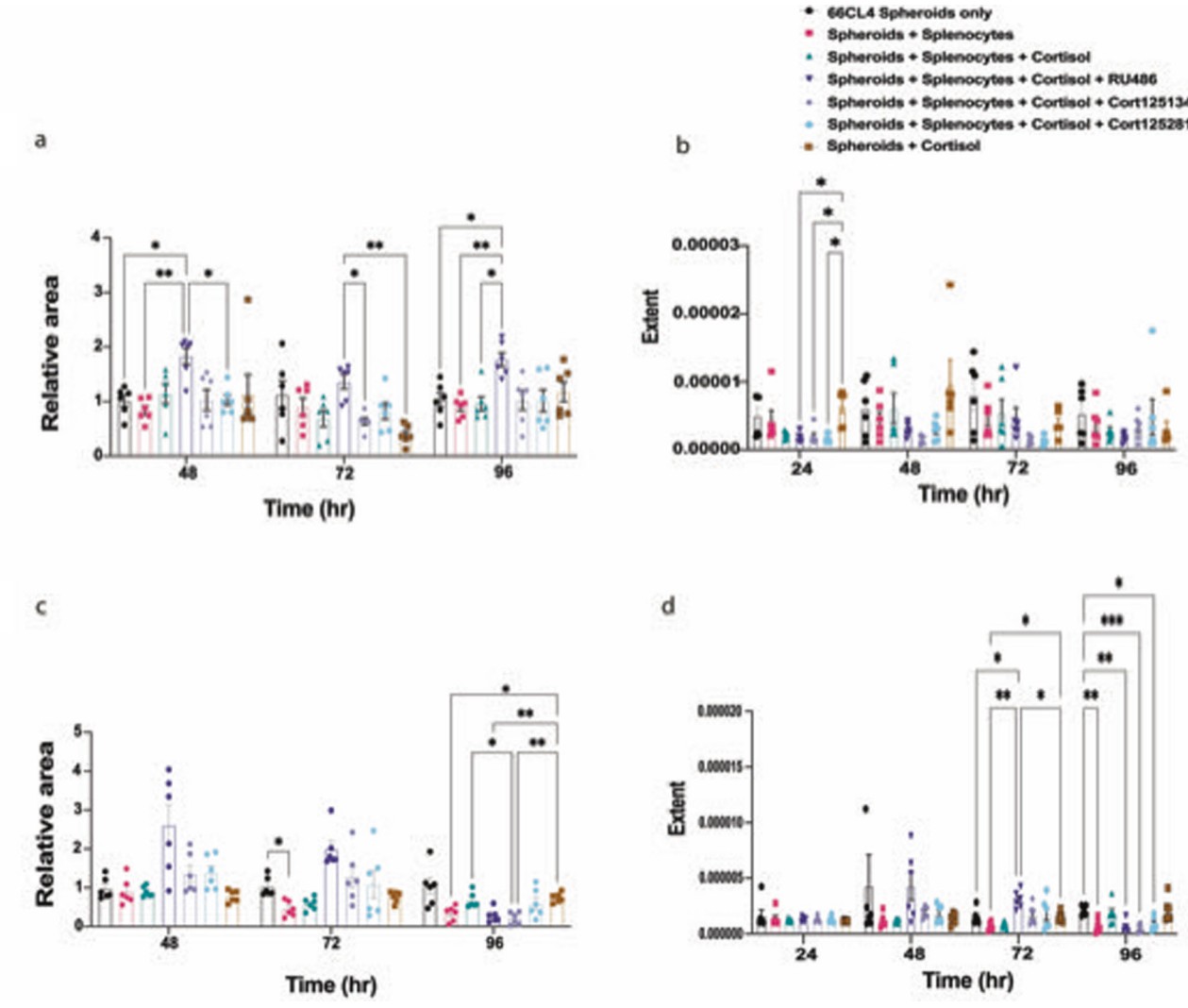

**Fig. 7 Spheroids' area and extent relative to immune infiltration and/or glucocorticoid receptor antagonists. a** 66CL4 spheroid relative area over time from different treatments **b** 66CL4 spheroids' roundness, by decreasing the extent, the treatment can have the tendency to disrupt the tumour spheroid shape **c** 4T1 spheroid relative area over time from different treatments **d** 4T1 spheroids' roundness, by decreasing the extent, the treatment can have the tendency to disrupt the tumour spheroid shape. Significant difference was determined by a two-way ANOVA (treatment + time) followed by Bonferroni post hoc test. Data represent mean ± SEM, $n = 6$ biologically independent spheroids *$p < 0.05$, **$p < 0.01$, ***$p < 0.001$.

not as complex (e.g., 2D). However, there are some considerations that need to be taken into account with our model. First, spheroid size is important and if the spheroid size gets too small then the infiltration reading can be misleading. For example, in the 4T1 cell line, spheroids were significantly smaller in the presence of glucocorticoid receptor antagonists (around 20% of the initial spheroid size). Ideally, a threshold should be set for spheroid size. The spheroid growth should also be monitored as the larger the spheroids' diameter, the more prominent the necrotic region[37]. In addition, it has been shown that with bigger spheroids, there will be more nutrient-poor and metabolically inactive layers[38].

Second, the type of spheroid formed is important. As described here, although we used two breast cancer cell lines, we observed different-shaped spheroids so an optimisation for each new cell line is recommended. Once the cell line is optimised and the type of spheroid is defined, the image analysis becomes easier to interpret as the TIS analysis works well for big round spheroids (e.g., 66CL4) while the TIC analysis works better for smaller loosely intact spheroids (e.g., 4T1).

Although the two image analysis approaches are based on different techniques (segmentation vs. machine learning), both TIS and TIC provide qualitatively, and quantitatively similar measures of trafficking as illustrated in Fig. 5. Both approaches measure the level of immune cell infiltration of the spheroid, which we interpret as the number of immune cells within the tumour spheroid boundary. Since to the best of the authors' knowledge, there is no established quantitative methodology for assessing infiltration that we could use to measure as well as to validate our proposed approaches, we have developed two independent approaches to be able to validate the approaches themselves implicitly, through the model comparison technique. Hence, we have used the two methods as a check for quantitative agreement.

In summary, the 3D co-culture model resembles in vivo settings, highlights the importance of the integration between bespoke image analysis algorithms and 3D in vitro cancer models, and paves the way for assessing the role of the immune system in cancer, analysis of immune-mediated mechanisms, and screening of novel drugs in vitro.

## Methods

**Cell lines**. Two murine triple-negative breast cancer cell lines were used, the 66CL4 and 4T1. The 66CL4 cell line was kindly provided by Dr Erica Sloane (Monash University, Australia), and was grown in Gibco Minimum Essential Media (MEM, Gibco) supplemented with 10% of foetal bovine serum (FBS, Gibco). The 4T1 cell line was obtained from ATCC and was grown in Gibco RPMI media supplemented with 10% FBS. All cells were cultured at 37 °C and 5% $CO_2$. 4T1 cells were purchased directly from ATCC, where they were authenticated. Cells were used within 10–15 passages of purchase. The mouse cell line 66CL4 was not authenticated as STR profiling is currently unavailable. All cell lines tested negative for mycoplasma by PI staining.

**3D spheroid formation**. When cells reached 70% confluency, they were scraped using a cell scraper and centrifuged at 300 $g$ for 5 min. The supernatant was discarded, and the cell pellet was resuspended in serum-free media and stained with a lipophilic fluorescent dye (Dio7778, Invitrogen) and incubated for 2 h. The cells were then centrifuged at 300 $g$ for 10 min, washed with PBS and centrifuged again at 300 $g$ for 10 min. The supernatant was discarded, and the pellet was resuspended with full media and spheroids were generated using Conning Coaster 96 ultra-low-bottom flask (Sigma–Aldrich, UK). In order to find the optimal seeding concentration of spheroids, a serial dilution of each cell line was resuspended in a volume of 30 µl of full medium per well. For the 66CL4 cell line, the number of cells to seed per well ranged from 200 to 2000. While for the 4T1, the number of cells to seed per well ranged from 250 to 2500. The plate was then centrifuged at 125 $g$ for 10 min.

**Viability assay**. The established spheroids were assessed for proliferation by the continuous MT Titre-Glo 3D cell viability assay (Promega, UK) using the manufacturer's instructions for cancer spheroids. After seeding the spheroids and incubating them in 37 °C and 5% $CO_2$, an equal amount of the viability reagent was added to the spheroids and incubated for 30 min on a shaker. Luminescence was quantified using a plate reader daily over 4 days. Luminescence was normalised as a percentage of the initial luminescence readout.

**Animals**. In total 10–12-week-old, female BALB/c mice, weighing 18 ± 2 g, were purchased from Charles River laboratories (Margate, Kent, United Kingdom). Mice were allowed to rest for one week after transportation and handled for a week prior to the start of the experiment so that they could acclimatise. Mice were kept at consistent conditions; temperature of 20–24 °C, humidity of 40–60% and a 12-h light/dark cycle. After the handling week, mice were euthanized to extract their spleens.

**T-cell activation**. Spleens were mashed through a strainer to isolate spleen cells (splenocytes). Cells were then incubated for 3 hr with a mixture of 1 µg/ml Ionomycin calcium salt from Streptomyces conglobatus (56092-82-1, Sigma–Aldrich) and 1 ng/m phorbol 12-myristate 13-acetate (16561-29-8, Sigma-aldrich) to activate cells from naive T to mature T cells.

**Flow cytometry**. The fresh isolated and activated splenocytes were stained by the murine PE-labelled anti-*CD3* antibody at a concentration of 0.5 mg/ml and the APC-labelled anti-*CD69* antibody at a concentration of 0.2 mg/ml (Biolegend, UK), and the viability stain 7AAD (Miltenyi, UK). Splenocytes were stained at a concentration of $1 \times 10^6$ cells/ml for each sample. Unstained cells, anti-*CD3* single-stained cells, anti-*CD69* single-stained cells, and single-stained cells with the viability dye- 7AAD were used as controls for this experiment. After staining, all samples were incubated for 30 min. in a cool dark place. Then, samples were washed twice and resuspended in FACS buffer (1% of bovine serum albumin in phosphate buffered saline). Finally, the samples were analysed using the BD Accuri c6 flow cytometer (BD Bioscience, UK) and the data were read, gated and analysed using the Flowjo software, version 10 (Flowjo, LLC, USA).

**In vivo study**. Female BALB/c mice (12 weeks of age) were injected with $1 \times 10^4$ 66CL4 cells/50 µL of PBS into the left, fourth mammary fat pad. The 66CL4 syngeneic mouse model was described previously[39]. Tumour development was initiated in two weeks, with tumour volumes of approximately 50–70 mm³. Tumours were measured weekly using a digital calliper and the tumour volumes were calculated using the formula: Vol (mm³) = $L \times W^2/2$, in which L is length (mm) and W is width (mm).

Two weeks post injection, mice were randomised into either a stress group ($n = 6$) or a nonstress group ($n = 6$). Mice from the stress group were placed individually in adequately ventilated tubes for 2 h on a daily basis for two weeks or until the tumour sizes exceeded 200(mm³)[40,41]. All primary tumours were harvested at necropsy, weighed and paraffin embedded for further analysis.

**Immunohistochemistry**. Paraffin-embedded breast tumour tissue was cut into 5-µm-thick transverse sections. Sections were placed in a 56 °C oven overnight, followed by deparaffination and rehydration using xylene and serial ethanol. Antigen retrieval was performed using Tris/EDTA buffer (Tris 10 mM, EDTA 1 mM, and Tween20 0.05%, pH9) at 95 °C for 20 minutes. Sections were rinsed with 1X PBS and blocked using 5% bovine serum albumin (BSA, Sigma) in 1X PBS for 1 h. The primary antibody against T-cell marker *CD3* (Rabbit anti-*CD3*ε, Cell signalling, #99940) was diluted in 2% BSA (1:150) and used to incubate the sections overnight at 4 °C. After thorough washing in 1X PBS sections were incubated with the secondary antibody (Anti-Rabbit IgG Alexa Fluor®488, ABCAM, ab150065) diluted in 2% BSA (1:2000) at 22 ± 2 °C for 4 hr in the dark, followed by DAPI (4′,6-diamidino-2-phenylindole, Sigma, D9542) counterstaining (1:10,000 in ddH₂O) at 22 ± 2 °C for 10 min. in the dark. Breast tumour tissue sections were coverslipped using ProLong® Gold antifade mountant (ThermoFisher, P36934).

Staining was imaged with confocal microscopy using Zeiss LSM 800, Axiocam 506, 40 × magnification. For quantitative fluorescence intensity analysis, all images were captured in a single uninterrupted run and uniform microscope settings were maintained. The images were exported as 8-bit tiff. files for *CD3*-positive cell quantification.

**Cell profiler analysis**. Analysis of *CD3* expression in breast tumour tissue from stressed and nonstressed mice was performed using the open-source image analysis software, CellProfiler (http://cellprofiler.org). Briefly, nuclei were automatically detected using the DAPI nuclear staining as a marker of the nuclear region and total number of nuclei were counted. Then *CD3*-positive cells were detected and counted. Specifically, the following sequence of commands was used: *Import images > Convert Colour to Grey > Identify Primary Objects (DAPI) > Identify Primary Objects (CD3) > Relate Objects > Export to Spreadsheet*. Then the percentage of *CD3*-positive cells was calculated by dividing the number of *CD3*-positive cells by the total number of cells, multiplied by 100 (%*CD3*-positive cells/total nuclei).

**3D co-culture**. Splenocytes, containing activated T cells, were co-cultured with 66CL4 spheroids. The splenocytes were stained with Dil D7777 stain lipophilic tracer (D282, Invitrogen). Prior to the actual co-culture, an optimisation experiment was performed, where spheroids and splenocytes were cocultured in three different ratios; 1:1, 1:3, and 1:5. The experiment was done at a ratio of 1:5. The co-culture was treated daily with corticosterone (1 µM) and the coculture was imaged daily using confocal microscopy. (Leica, Germany). Spheroids were later co-cultured and split into 7 groups: (1) spheroids only, (2) spheroids and splenocytes, (3) spheroids, splenocytes and corticosterone (1 µM) (4) spheroids, splenocytes, corticosterone and RU486 (1 µM), (5) spheroids, splenocytes, corticosterone, and CORT125134 (1 µM), (6) spheroids, splenocytes, corticosterone, and CORT125281 (1 µM), and (7) spheroids and corticosterone. Spheroids from each group were treated daily for 4 days, and the media was saved for further analysis.

**Confocal imaging**. Each spheroid was imaged daily using confocal microscopy (Leica, Germany). The laser used to excite the DiO 7778 was set at 488 using a filter for the emission between 485 and 550 nm, while for the Dil D7777 the laser was set at 543 nm using a filter for the emission between 550 and 600 nm. Magnification was set at × 10 and × 20. Images that were acquired at maximum projection setting at different Z values, focusing the plane at the spheroid level, were used for the mathematical analysis.

**Cytokine detection**. Media collected from the co-culture were collected to measure the concentration of cytokines IFN-γ (900-M98, Peprotech, UK) and IL-10 (900-M53, Peprotech, UK) where each sample had three technical and three biological replicates. The concentration of the cytokine was determined using the ABTS ELISA kit (900-K00, Peprotech, UK). The experiment was conducted following the manufacturer's instructions. The release of both IFN-γ and IL-10 was expressed as pg/ml.

**Normalised changes of spheroid areas**. To analyse the changes in the spheroids from the experiments, we computed their areas and normalised them against their corresponding initial figures at the 24th h. First, we separated the original images via the RGB colour map and took the green channel images. Second, to reduce noise, we employed the standard 2-D median filter[42], and applied 200 iterations. The results are segmented by the Otsu's method[43], which selects a global threshold value by minimising intraclass intensity variance of the images' histograms. Third, we counted the pixels that have nonzero intensity values to produce the spheroid area in that image. Finally, the results from each set were normalised by dividing by the corresponding area figures at the 24th h. Therefore, the normalised changes at 48th, 72nd, and 96th h may be reported.

**Extent between spheroid area and its minimal bounding box**. Using the spheroid areas, we calculated the so-called extent. This is the ratio between the spheroid area and the size of its minimal bounding box (i.e., the smallest rectangular area containing all spheroid(s)). We calculated the size of the minimal bounding box for the spheroids within each image, using the MATLAB Image Processing Toolbox[44]. The extent was computed by taking the spheroid areas and dividing by the corresponding size of the minimal bounding boxes.

**Segmentation-based trafficking index: TIS**. TIS: This is a two-stage approach where we first identify the tumour boundary using image segmentation techniques, and second, we measure the ratio of red over green channel intensity within the tumour in an attempt to control the spurious (red channel) signal arising from the tumour cells that exhibit green channel fluorescence.

Given an image, we assumed that the cancer cells and immune cells appeared in tumour spheroids and immune cell regions defined as one or several connected regions of a specific colour (green and red, respectively, in the present study). Methodically, we viewed both the tumour masses and the immune cells as one or several closed target areas in two dimensions (in the present study we only deal with planar images although our methodology immediately can be extended to 3d data). Initially, we took an image from the experiment and separated the red and green channels from its RGB colour map into two different grey images, where the grey intensities in each grey image represent the amount of red and the green colour from the original image, respectively. Second, we identified the closed target areas in each of the grey images via a segmentation as described above (area). Using these segmented sets, we construct a segmentation-based trafficking index (TIS), which is a dimensionless number that is the sum over all pixels that are in both segmented regions of the red intensity divided by green intensity normalised by the total number of pixels in the set. Taking this ratio allowed us to account (at least partially) for the fact that there is a level of red intensity associated with tumour cells.

**K-means classification-based trafficking index (TIC)**. TIC: This is also a two-stage approach in which we first use a machine learning algorithm to classify each pixel in the image into different groups obtaining four classes, background, tumour, immune cell or colour saturated. The identification of the colour saturated class allows the exclusion of pixels that cannot be quantified appropriately, since the colour saturation hinders any comparison between the red and green levels. The trafficking measure is then based on the number of pixels classified as immune cells that are completely surrounded by pixels classified as tumour cells.

In order to obtain an independent quantification of the level of trafficking, we applied a k-means algorithm[26] to classify the pixels in the images as background, cancer cells, immune cells or colour-saturated. K-means classification finds the best classification of data points in a given number of classes, by minimising the intraclass variance. The k-means algorithm was implemented in-house using Python[45]. We created a training dataset by randomly selecting one image from each group and time point, for a total of 28 images. We then applied k-means clustering with four categories. Although we expected three main classes in the images—background, cancer spheroid, and immune cells—we included a fourth class that allowed us to distinguish noncharacteristic pixels, usually corresponding to saturated colour values. Note that in a saturated pixel, we lost some information about the actual cell densities (intensity level) in the image and therefore the data cannot be interpreted with certainty, moreover as described above this allows us to account (at least partially) for the fact that there is a level of red intensity associated with tumour cells. We ran the algorithm until the difference between consecutive iterates was less than $10^{-6}$. We validated the results by repeating the same procedure using different training datasets, selected as described above. In all cases, the algorithm converged to the same classes, in less than 30 iterations. We used the classifier to quantify the pixel types in the images. In order to measure the number of immune cells within the spheroid, we identified the class of the neighbourhood of a given pixel as the class of the majority of the pixels in the surrounding $11 \times 11$ square. We then quantified trafficking by computing the total number of immune cell-surrounded-by-cancer pixels, divided by the sum of the number of immune-surrounded-by-cancer and cancer-surrounded-by-cancer pixels.

**Statistics and reproducibility**. Data were first tested for normality of distribution. Data that were normally distributed were analysed by a one/two-way ANOVA (depending on the number of independent variables) followed by a post hoc test to identify differences between groups. Otherwise, when data that were not normally distributed, they were analysed using a nonparametric Kruskal–Wallis test followed by a post hoc Dunn's multiple-comparison test. For experiments on spheroids, six biological repeats were used. The statistical test and n number is stated in each figure legend.

**Reporting summary**. Further information on research design is available in the Nature Research Reporting Summary linked to this article.

## Data availability

All data generated computationally are included. Campillo-Funollet, Eduard, & Yang, FengWei. (2021, April 26). ecam85/spheroids: Spheroid image analysis (Version v1.0). Zenodo. https://doi.org/10.5281/zenodo.4719445 or https://github.com/ecam85/spheroids. All other raw data can be obtained from the corresponding authors on reasonable request.

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

## Acknowledgements

This work (AM) was supported by an individual grant from the Dr Perry James (Jim) Browne Research Centre on Mathematics and its Applications (University of Sussex). AM's work was partially funded by grants from the European Union Horizon 2020 research and innovation programme under the Marie Sklodowska–Curie grant agreement (No 642866), the EPSRC (EP/J016780/1, EP/T00410X/1), the Health Foundation (1902431), the NIHR (NIHR133761), and the Leverhulme Trust Research Project Grant (RPG-2014-149). AM is a Royal Society Wolfson Research Merit Award Holder funded generously by the Wolfson Foundation. AM is a Distinguished Visiting Scholar to the Universita degli Studi di Bari Aldo Moro, Bari, Italy and the Department of Mathematics, University of Johannesburg, South Africa. FWY thanks EPSRC(EP/R001588/1) for their funding and support and MF thanks The Boltini tTrust for their funding.

## Author contributions

M. F. and G. A. conceived the idea and designed the experiments. G. A., M. M., M. F. and H. I. carried out all the lab experiments. A. M., F. W. Y., E. C. F. and C. V. conducted all the framework analysis. H. H. and A. G. contributed to the receptor antagonists and writing the GR results section, and M. F., G. A., and C. V. wrote the paper. A. M. edited the paper. All authors contributed to writing sections of the paper.

## Competing interests

All authors have no competing interest, except that Hazel Hunt and Andrew Greenstein are both Corcept stockholders.

### Ethics statement

The experiments involving animals and Schedule 1 protocols used to obtain breast tumour tissue were approved by the Animal Welfare and Ethical Review Body (AWERB committee) of the University of Brighton. This study was conducted in accordance with the principles of the Basel Declaration and adhered to the legislation detailed in the United Kingdom Animals (Scientific Procedures) Act 1986 Amendment Regulations (SI 2012/3039). All efforts were taken to maximise animal welfare conditions and to reduce the number of animals used in accordance with the European Communities Council Directive of September 20th, 2010 (2010/63/EU).
