## [Peer Review File · Communications Biology]

Reviewers' Comments:

Reviewer #1:

Remarks to the Author:

This paper aims to develop and apply methods for assessing infiltration of lymphocytes in co-culture models. Reliable methods for quantifying images in this way are needed by the research community. While this paper has many interesting aspects, in my view there are some points that need to be addressed prior to publication:

- Figure 1. Celltitre glow data for the spheroids are shown at a single timepoint. The numerical value is not related to anything (e.g. a standard curve so that cell number can be evaluated). It would be more valuable to see viability over time or if Live/Dead staining were used to assess proportion of viable cells. Viability at a single timepoint is meaningless.
- Although seeding density was varied in expts shown in Figure 1, it is not clear what seeding density was used subsequently
- Similarly, and more importantly, although multiple timepoints are shown in Figure 1, it is not clear at what timepoint the splenocytes were added to form the coculture. Based on the information provided, it seems that the splenocytes were added right at the beginning of the experiment. If so, it is not infiltration that is being measured as the spheroids are not fully formed at that point and the splenocytes are simply being allowed to integrate into the forming spheroid. Consequently, any subsequent quantification of the numbers of splenocytes present is a measure of retention/loss, not infiltration.
- The CD3/CD69 phenotypes of the splenocytes activated for 24hrs should be included
- The uptake and retention of the dye in the different types of splenocytes used (activated, not activated, and for later experiments, following treatment with cortisol and antagonists) should be assessed over time – it may not be constant and may vary under different conditions. This might affect quantification and interpretation of the signals arising from these cells.
- Figure 3 – although tumour volumes were not statistically significantly different, it should be acknowledged in the text that there was some change
- It would have been helpful in the experiments done to validate the in vivo experiments in the spheroid co-cultures if the same methods had been applied e.g. some immunostaining in spheroid sections.
- The value in the two different approaches for assessing infiltration should be clarified. This could perhaps be achieved by explaining them more clearly, without using technical terms (e.g. 'the tumour masses are segmented in each image' or 'The TIS is computed as the sum over each pixel' which are not clear for the non-expert) i.e. what does each approach actually measure in practical terms. In the discussion, it would be helpful if the two approaches could be compared and contrasted and advice given as to when to use each.
- Also, in relation to this, later on, why is only TIS shown in Figure 6, although both are mentioned in the text. From my understanding, the TIC seems more sophisticated as it is based on immune cells being surrounded rather than in proximity to tumour cells, so I am not clear why it was not used in this analysis.
- The data in Figure 5 is mentioned as validating the 3D co-culture model based on seeing a similar pattern to that seen in the in vivo experiment. However, no evidence is provided that this similarity in pattern is due to the same underlying processes and this would really be needed to make this claim. I would have preferred to see some higher powered images of areas of the spheroids and associated quantifications to be convinced that the quantification methods were working.
- The quantification does not always seem to correlate with what is seen in the images. E.g in Figure 6, the untreated and Cortisol+125134 antagonist-treated 66CL4 co-cultures look quite similar, with a lot of signal at 96 hrs yet the quantification of the infiltration is very different for these conditions. And in some cases, at early timepoints, the spheroids look completely green, yet at later timepoints there is yellow or red signal. There should be explanation/discussion of this. Again, it would help if some example high-powered images were provided, including illustrations of red/green/merged images.
- Considering all of the above points that could be clarified and discussed, the Discussion is very brief and repetitive of points already made in the Results.

Minor points:

- In a number of places, 'however' is incorrectly punctuated. It should be preceded by ; and followed by , when used to join two independent clauses e.g. end of first paragraph on page 2.
- Figure 1 A1-A6: these labels are really hard to see

Reviewer #2:

Remarks to the Author:

This paper uses novel methods to track and monitor immune cell infiltration into breast cancer spheroids in vitro, and then looks at interaction with the cancer cells. For this a method of tagging with lipophilic tracers was used. Further work was done in vivo and the methods were extended to study the efficacy of glucocorticoid receptor antagonists.

This is a clear and concise piece of work, however there are some issues. there is potential in this work- but more needs to be done to take certain aspects forward.

1) The data around the activated and inactivated splenocytes (specifically the CD69 (+)ve cells) is difficult to interpret. The inactivated cells were "not easily detected". Does that mean they didn't infiltrate? or they lost the tracer?

2) All the work around the glucocorticoid receptor antagonists appears to show trends, but no real significance. So how can this method of be of real use with significant outcomes reported? Particularly this is around figure 6.

3) Figure 3 - staining of CD3 needs to be clearer.

4) There is no mention of the heterogenous viability of cells within the spheroids of different sizes. for this kind of method an understanding of penetration depth of the stain is also critical, and the potential cell death in the core of spheroids also necessary information.

I would propose this is a more specialised methods paper for a speciality journal.

Dear Dr. Karli Montague-Cardoso and reviewers,

We thank you for the positive review of our manuscript. We have taken each comment and suggestion on board which taken together have significantly improved our manuscript. More specifically, we have performed additional experiments including; viability experiments, adding higher power images, CD3/CD69 phenotypes of the splenocytes activated for 24 hours and inclusion of characterisation of the uptake and retention of dye of the different splenocytes. We have written a line by line rebuttal and highlighted changes in the text of the manuscript to make it easier for review.

Reviewer #1

1. Figure 1. Celltitre glow data for the spheroids are shown at a single timepoint. The numerical value is not related to anything (e.g. a standard curve so that cell number can be evaluated). It would be more valuable to see viability over time or if Live/Dead staining were used to assess proportion of viable cells. Viability at a single timepoint is meaningless.

Authors response: We thank the reviewer for this positive comment. We have conducted an experiment illustrating viability over time from 24-96 hours. The cells remain viable for up to 96 hours (Figure 1C and descriptive text on page 4).

2. Although seeding density was varied in expts shown in Figure 1, it is not clear what seeding density was used subsequently

Author's response: We apologise for the oversight. All experiments for 66CL4 spheroids were seeded at 1000 cells/well and for 4T1 with 1500 cells/well, respectively. We have amended the descriptive text (page 4) and figure legend for Figure 1B to highlight the seeding densities used.

3. Similarly, and more importantly, although multiple timepoints are shown in Figure 1, it is not clear at what timepoint the splenocytes were added to form the coculture. Based on the information provided, it seems that the splenocytes were added right at the beginning of the experiment. If so, it is not infiltration that is being measured as the spheroids are not fully formed at that point and the splenocytes are simply being allowed to integrate into the forming spheroid. Consequently, any subsequent quantification of the numbers of splenocytes present is a measure of retention/loss, not infiltration.

Authors response: We apologise for not making it clear at what timepoint the splenocytes are added to form the coculture. Spheroids were grown first for 4 days before addition of immune cells. We have now included a timeline that demonstrates infiltration as measured in our manuscript (Figure 5A).

4. The CD3/CD69 phenotypes of the splenocytes activated for 24hrs should be included

Authors response: We have now included this data (Figure 2) and added appropriate text in the results section (page 6).

5. The uptake and retention of the dye in the different types of splenocytes used (activated, not activated, and for later experiments, following treatment with cortisol and antagonists) should be assessed over time – it may not be constant and may vary under different conditions. This might affect quantification and interpretation of the signals arising from these cells.

Authors response: We agree with the reviewer and have included an extra experiment to support quantification and interpretation of Immune cell infiltration and amended our descriptive text (page 14) and Supplementary Figure 3.

6. Figure 3 – although tumour volumes were not statistically significantly different, it should be acknowledged in the text that there was some change

Authors response: We agree with the reviewer. We have amended the sentence in the results section and it now reads ‘As expected, and although stressed tumours appeared smaller, primary tumour weight and volume were not significantly different between stressed and non-stressed groups’ (page 8).

7. It would have been helpful in the experiments done to validate the in vivo experiments in the spheroid co-cultures if the same methods had been applied e.g. some immunostaining in spheroid sections.

Authors response: This is a good point and we considered this. However, it is extremely challenging to do IHC on our spheroids due to their small size, especially after the infiltration of immune cells. This limitation is discussed in the discussion (page 20).

8. The value in the two different approaches for assessing infiltration should be clarified. This could perhaps be achieved by explaining them more clearly, without using technical terms (e.g. ‘the tumour masses are segmented in each image’ or ‘The TIS is computed as the sum over each pixel’ which are not clear for the non-expert) i.e. what does each approach actually measure in practical terms. In the discussion, it would be helpful if the two approaches could be compared and contrasted and advice given as to when to use each. Also, in relation to this, later on, why is only TIS shown in Figure 6, although both are mentioned in the text. From my understanding, the TIC seems more sophisticated as it is based on immune cells being surrounded rather than in proximity to tumour cells, so I am not clear why it was not used in this analysis.

Authors response: The reviewer raised an important point which we are very happy to address to strengthen the usefulness of the image analysis techniques. Both TIC and TIS image analysis approaches seek to measure the level of immune cell infiltration into the spheroid, which we interpret as the number of immune cells (corresponding to red channel intensity) within the

tumour spheroid boundary. To the best of the authors' knowledge, there is no established quantitative methodology for assessing infiltration, hence we developed two independent approaches to allow for comparison and validation.

A challenge that must be addressed in computing both statistics is that tumour cells themselves express red channel intensity giving rise to spurious evidence of infiltration. The non-technical summary of both approaches included below has been added to our manuscript in the Methods section.

"In practical terms, we summarise the two image analysis approaches in the methods section pages 27- 28 and as follows:

TIS: This is a two-stage approach where we first identify the tumour boundary using image segmentation techniques, and second, we measure the ratio of red over green channel intensity within the tumour to control for the spurious (red channel) signal arising from the tumour cells which exhibit green channel fluorescence.

TIC: This is also a two-stage approach in which we first use a machine learning algorithm to classify each pixel in the image into different groups obtaining four classes, background, tumour, immune cell or colour saturated. The identification of the colour saturated class allows the exclusion of pixels that cannot be quantified appropriately, since the colour saturation hinders any comparison between the red and green levels. The trafficking measure is then based on the number of pixels classified as immune cells which are completely surrounded by pixels classified as tumour cells.

We have included the results of both approaches in Figure 6. Although not directly numerically comparable, both approaches broadly agree in terms of significance tests. We have added some discussion around areas of similarity and differences between the approaches to the conclusion of the manuscript.

9. The data in Figure 5 is mentioned as validating the 3D co-culture model based on seeing a similar pattern to that seen in the in vivo experiment. However, no evidence is provided that this similarity in pattern is due to the same underlying processes and this would really be needed to make this claim. I would have preferred to see some higher powered images of areas of the spheroids and associated quantifications to be convinced that the quantification methods were working.

Authors response: This is an important point made by the reviewer. We have now included high power images of the Figures (Supplementary Figure 2) and their corresponding analyses. We found that the results from high powered images were comparable to the low power images. However, higher magnification can increase light scattering which is considered an inherent limitation of confocal imaging of thick specimens (page 19).

10. The quantification does not always seem to correlate with what is seen in the images. E.g in Figure 6, the untreated and Cortisol+125134 antagonist-treated 66CL4 co-cultures look

quite similar, with a lot of signal at 96 hrs yet the quantification of the infiltration is very different for these conditions. And in some cases, at early timepoints, the spheroids look completely green, yet at later timepoints there is yellow or red signal. There should be explanation/discussion of this. Again, it would help if some example high-powered images were provided, including illustrations of red/green/merged images.

Authors response: Regarding the example mentioned, the infiltration of the CORT+125134 treated cells is higher than cells treated with Cortisol alone, as observed both in the images and on the graphs, but still lower than that of spheroids and splenocytes, as observed in the images and graphs. At early time points the spheroid is green because immune cells need time to infiltrate into the spheroid and one can see green and red on the same object when they infiltrate. In some cases, one can see yellow due to the overlap between red and green on the same place. Please note that before the image analysis is conducted, images are split into red only and green only.

11. Considering all of the above points that could be clarified and discussed, the Discussion is very brief and repetitive of points already made in the Results.

Authors response: Thank you for pointing this out. We agree with the reviewer and we have now edited our discussion to remove any repetitive text and replaced discussion of the reviewers and editor's salient points.

12. In a number of places, 'however' is incorrectly punctuated. It should be preceded by ; and followed by , when used to join two independent clauses e.g. end of first paragraph on page 2.

Authors response: Corrected

13. Figure 1 A1-A6: these labels are really hard to see

Authors response: Corrected

Reviewer #2

1) The data around the activated and inactivated splenocytes (specifically the CD69 (+)ve cells) is difficult to interpret. The inactivated cells were "not easily detected". Does that mean they didn't infiltrate? or they lost the tracer?

Authors response: We apologise for the misunderstanding, to clarify the inactivated T cells did not infiltrate the tumour effectively—we have amended the text in the results section on page 6. We have also added another timepoint of 24 hours for clarity (Figure 2b).

2) All the work around the glucocorticoid receptor antagonists appears to show trends, but no real significance. So how can this method of be of real use with significant outcomes reported? Particularly this is around figure 6.

Authors response: Initially, we did not design this part of the paper as a powered study because we were simply examining patterns between the images and quantification using our 2 methods. However, to improve and strengthen our study we have conducted a stringent statistical analysis and found that receptor antagonists had a significant effect on the size and roundness of the spheroids (page 14-15). We have clarified *p* values in the text as it over complicates the graphs. In addition, both algorithms agree that CORT125281 decreased infiltration on 66CL4 spheroids (Figure 6). We have also explained the limitations of the technique when the spheroids get smaller when treated with the antagonists (discussion Page 19).

3) Figure 3 - staining of CD3 needs to be clearer.

Authors response: We used IF to stain mouse tissue and similar staining was obtained compared with other IF in mouse tissue (Parra E, Uraoka N, Jiang M, Cook P, Gibbons D, Forget M et al. Validation of multiplex immunofluorescence panels using multispectral microscopy for immune-profiling of formalin-fixed and paraffin-embedded human tumour tissues. Scientific Reports. 2017;7(1)).

4) There is no mention of the heterogenous viability of cells within the spheroids of different sizes. for this kind of method an understanding of penetration depth of the stain is also critical, and the potential cell death in the core of spheroids also necessary information.

Authors response: This is a good point raised by the reviewer. We have assessed viability over time and provided high power images to highlight the depth of the stain for example in Figure 5B you can see the necrotic core and high penetration of dye into the spheroids.

Reviewers' Comments:

Reviewer #1:

Remarks to the Author:

Thanks to the authors for taking on board the reviewers' comments constructively. These have now been addressed to my satisfaction.

Reviewer #2:

Remarks to the Author:

This is a very interesting paper looking at the development of algorithms for the accurate quantification of image analysis, specifically looking at the infiltration of immune cells into spheroids.

I believe the science is sound, the authors have shown that they can monitor the penetration of a distinct infiltrating cell population into a spheroid of a different cell population.

Although this is very good- I am interested in how this image analysis can be linked to specific biological phenomena. It is interesting to note the heterogeneity of cell presence mainly in the spheroid periphery. Whilst it is well known that cells in the core of spheroids die due to a lack of oxygen and nutrients (core necrosis), it would be interesting if algorithms were developed to study hypoxia gradients, metabolic gradients and the like. Maybe the authors could discuss this.

We would like to thank the reviewers for their positive reviews of our manuscript.

Reviewer comment It is interesting to note the heterogeneity of cell presence mainly in the spheroid periphery. Whilst it is well known that cells in the core of spheroids die due to a lack of oxygen and nutrients (core necrosis), it would be interesting if algorithms were developed to study hypoxia gradients, metabolic gradients and the like.

Authors response: We agree with the reviewer and we have included the following sentences in our discussion highlighting work of others on this topic: 'The spheroid size should also be monitored as the larger the spheroids' diameter, the more prominent the necrotic region⁴⁴. In addition, it has been shown that with bigger spheroids, there will be more nutrient-poor and metabolically inactive layers⁴⁵'.

1) Please ensure that you upload the attached checklist when submitting your revision

completed

2) When addressing the final point of reviewer 2, please state in your cover letter how you have addressed this rather than showing tracked changes in the resubmitted manuscript.

completed

3) Please do not have a border around your author affiliations and please place these directly below the author list.

Completed

4) Please check the attached document for guidance on affiliation numbering and author emails

Completed

5) Please do not embed figure legends in with your figures.

Completed

6) Please ensure that all supplementary figures and legends are only in a single supplementary PDF

Completed

7) Please ensure that all full blots/gels are provided in the supplementary PDF and that they are labelled with the figure that they correspond to.

completed

8) Please add data points to all of your graphs with error bars.

Completed

9) We recommend that FACS gating is included in the supplementary PDF.

Completed

10) Please ensure that the scale bar is clearly visible for all images.

Completed